# Improving the management of musculoskeletal conditions: can an alternative approach to referral management underpinned by quality improvement and behavioural change theories offer a solution and a better patient experience? A mixed-methods study

Victoria Tzortziou Brown,[1] Martin Underwood,[2,3] Olwyn M Westwood,[4] Dylan Morrissey[1]

For numbered affiliations see end of article.

**Correspondence to**
Dr Victoria Tzortziou Brown;
v.tzortzioubrown@qmul.ac.uk

## ABSTRACT

**Objectives** To assess whether a quality improvement-based approach to referral management can result in better musculoskeletal care within general practice.

**Design** Prospective cohort study using mixed methodology including random-effects meta-analysis and interrupted time series.

**Setting and participants** 36 general practices in East London.

**Intervention** Informed by the results of a Cochrane review on educational interventions to improve general practitioners' (GPs) musculoskeletal care, we developed a multifaceted intervention, underpinned by quality improvement and behavioural change theories. It combined locally agreed clinical pathways, feedback on referral rates, clinical audit and peer review.

**Main outcome measures** Referral letter content, pathway adherence, referral rates, inter-practice variability and patient experience were evaluated before and after the intervention.

**Results** Referral letter content on suspected diagnosis and prior management improved from a pooled preintervention proportion of 59% (95% CI 53% to 65%) and 67% (95% CI 61% to 73%), respectively, to 77% (95% CI 70% to 84%) and 81% (95% CI 74% to 88%). Pathway adherence improved from a pooled preintervention percentage of 42% (95% CI 35% to 48%) to 66% (95% CI 57% to 76%). The effect was greater across all quality outcomes for practices with baseline performance below or equal to the pooled baseline performance. There were reductions in the variability and rates of orthopaedic referrals at 6, 12 and 18 months (referral rate relative effect 32% (95% CI 14% to 48%), 30% (95% CI 7% to 53%) and 30% (95% CI 0% to 59%), respectively). Patient rating of how well GPs explained the musculoskeletal condition improved by 29% (95% CI 14%

### Strengths and limitations of this study

► The intervention was implemented within the constraints of clinical commissioning and rapid policy change demonstrating its feasibility and applicability within similar health systems.

► Unlike other healthcare interventions, it was developed with an underpinning behavioural theory and incorporated a number of behaviour influencing factors.

► The long follow-up on the impact on referral rates provided an opportunity to assess the sustainability of behavioural change over time.

► The evaluation incorporated a process evaluation and an outcomes evaluation which assessed the impact of the intervention across a range of outcomes including referral quality, referral rates, rate variability and patient experience.

► The intervention targeted behaviours at a number of levels and our evaluation could not identify the most cost-effective elements.

to 43%) and patient perception on the usefulness of the GP appointment improved by 24% (95% CI 9% to 38%).

**Conclusions** A quality improvement-based approach to referral management which values GPs' professionalism can result in improvements across a range of outcomes including referral quality, patient experience, referral rates and variability.

## INTRODUCTION

A strong primary care infrastructure is central for a cost-effective healthcare system.[1] The pivotal role of the general practitioner (GP)

in providing person-focused care incorporating the gatekeeper function between primary, and the high-cost and intervention-heavy specialist care, can be crucial for optimal and sustainable healthcare delivery especially at a time of increasing demand.[2–4]

Typically, 5% of GP consultations result in a specialist referral,[2 5] giving rise to approximately 13.6 million referrals in 2015–2016 in England, an increase of over 18% in 5 years.[6] There have been reports that up to 30%–40% of these may be clinically inappropriate or avoidable.[7 8] However, the definition of 'appropriateness' is neither closely nor reliably defined[9] reflecting the multiple factors affecting referral behaviour[10–15] and the lack of full understanding of the causes for the variation in referral rates.[13 16 17]

Furthermore, available evidence on the effectiveness of different referral management approaches is limited and conflicting, with similar approaches resulting in different outcomes when applied to different contexts.[15 18] The theoretical underpinning of such initiatives is often unclear, their main focus is increasingly on reducing referral numbers and the majority of the available studies have evaluated the effect of interventions on just one aspect of a range of potential outcomes.[15] However, intermediate outcomes, such as the content of the referral, are key important markers of change that should be evaluated in order to determine where blocks to, or facilitators of, system-wide impact may be occurring[15] and to better understand the mechanisms by which interventions influence behavioural change.[19]

The aim of this study was to evaluate the impact of a theory-based, quality improvement (QI) approach to GP musculoskeletal management and referral behaviour across a range of outcomes, including the quality of referral letters, the number and variability of referral rates and patient experience and by doing so, to address some of the evidence gaps in this important area.

## METHODS

### Study design

We evaluated the implementation of a general practice focused intervention grounded in evidence synthesis underpinned by behavioural change theory, based on QI principles and using mixed methodology. We used the Medical Research Council (MRC) framework for complex interventions[20] in the design, implementation, evaluation and piloting phases of the multifaceted intervention. The evaluation incorporated a process evaluation guided by HM Treasury's guidance for evaluation 'The Magenta Book',[21] and an outcomes evaluation which assessed the impact of the intervention on referral quality, referral rates, rate variability and patient experience.

### Setting and participants

The intervention took place in the London Borough of Tower Hamlets which has an ethnically diverse population with high levels of deprivation and a lower healthy life expectancy than the national average.[22]

Since 2009, the 36 general practices have been organised into eight networks each serving a population of approximately 30 000–40 000 patients. All practices were included in the intervention.

Apart from the trauma and orthopaedics (T&O) and rheumatology hospital outpatient service, people in Tower Hamlets also have access to a local intermediate musculoskeletal service (clinical assessment service).

### The intervention

The intervention had components targeting the main factors which influence clinical behaviour and behavioural change across the 12 domains described in the theoretical domains framework (TDF).[23] This framework has been used successfully in the design of interventions aiming to change health professionals' behaviour in a number of clinical settings, including primary care.[24–26] A detailed account of all the components of the intervention, how they link to the above TDF and the supporting theories which underpin them is presented in table 1 (table adapted from Table on Theoretical Support for Mental Health QuERI Antipsychotic Treatment Improvement Program Components and Tools[27]).

The intervention was part of the Tower Hamlets Clinical Commissioning Group (CCG) Network Improvement Scheme (NIS), which acted as the system driver providing the incentives for change while ensuring a rigorous, structured approach incorporating planned objectives within set timelines over a year period. The CCG's project management team ensured that the intervention would be delivered at scale and pace. The NIS offered allocated funding at the network level for the clinical audits, for hosting educational events and for clinician backfill to attend these. There were no financial incentives attached to clinical performance targets such as referral rates because such incentives may have a negative impact on the quality of care[28] and introduce conflicts of interests and ethical dilemmas.[29]

### Process evaluation

The process evaluation included an assessment of how well each of the components of the intervention was implemented, including the level of GP practice participation in the clinical audit, educational meetings and outreach visits. The assessment was based on data from CCG and provider progress reports. The degree of GP participation and the feedback and reflections of GP practices and networks in their reports were analysed and grouped thematically. The results provided a measure of participants' interaction with the intervention.

### Outcomes evaluation

#### Referral quality

To assess the impact on referral quality, we assessed three attributes as described by Blundell et al[30]: necessity, quality of process and destination. To determine necessity, we

**Table 1** Summary of intervention components and the theoretical framework underpinning them

| Intervention component and brief description | Theoretical domains framework factor(s) addressed | Supporting theories of behavioural change |
|---|---|---|
| Local consensus processes: Local clinical pathways on common musculoskeletal conditions (low back pain, shoulder pain and osteoarthritis) designed with input from multidisciplinary teams (using the Delphi technique) summarising guideline recommendations and availability of local services. These pathways were disseminated to all 36 practices, were presented at local educational events and were published on the local website for general practitioners (GPs) to access. | Knowledge, beliefs about consequences, memory and decision processes. | Theory of planned behaviour Bandura's social cognitive theory Diffusion of innovation. Rational system theory. |
| Clinical audit: Audit and reflection on current practice using standardised proformas, opportunity for team discussion of findings, identification of learning needs and agreeing ways to improve practice. The audits took place at practice level and the results were discussed both within practices and also at network events. | Knowledge, motivation and goals, social influences, behavioural regulation. | Theory of planned behaviour Bandura's social cognitive theory. Diffusion of innovation. Social influence theory. Complexity theory. Rational system theory. |
| Feedback: Monthly reports to provide ongoing feedback to clinicians on referral activity (comparative data at GP, practice, network and borough levels). These reports were disseminated both at practice and network levels. | Motivation and goals, social influences, behavioural regulation. | Theory of planned behaviour. Bandura's social cognitive theory. Diffusion of innovation. Social influence theory. Rational system theory. Statistical process control theory. |
| Monthly educational meetings and outreach visits facilitated via local opinion leaders: Opportunity to discuss the clinical pathways and clinical practice with peers (case-based discussions). Local opinion leaders (referral champions) in each network of practices facilitated discussions on referral decision-making, best clinical practice and identification of learning needs. The meetings were attended by GPs and practice managers from each practice. | Knowledge and skills, motivation and goals, social/professional role and identity, social influences, behavioural regulation. | Theory of planned behaviour Bandura's social cognitive theory. Diffusion of innovation. Social influence theory. Natural system theory. |

measured the adherence of referrals to the agreed local clinical pathways on back pain, shoulder or osteoarthritis (provided that the reason for the referral was one of the above conditions). The pathways covered all the aspects of management within primary care including the indications for investigations, medication, physiotherapy, the role of steroid injections and provided resources on exercise and advice for patients. The information on the referral letter and EMIS notes was compared with the relevant pathway during the audit process. The EMIS records were used in addition to the information contained in the referral letters in order to ensure that all relevant information on the prior management of patients was captured. We recognised that a simple yes/no answer could be subjective and might not reliably capture adherence to the clinical pathway. Therefore, the audit proforma incorporated questions on the specific management of the patient prior to the referral and on what could have

been done differently according to the pathway. We used the information contained in the referral letters as an indicator of the quality of process. Finally, we calculated surgical conversion rates and the percentage of referrals that were seen once and discharged from the hospital without intervention and used these as indicators of the appropriateness of referral destination.

### Data collection and analysis

Practices retrospectively audited consecutive musculoskeletal referrals before and after the intervention.

The data were presented descriptively and summarised using tables and charts. Additionally, random-effects meta-analysis was undertaken (using StatsDirect statistics software)[31] to estimate the relative risk (RR) of the intervention effect. Each practice was taken as an individual study case and the results meta-analysed to produce a composite RR. The baseline proportion was calculated

after pooling the proportions in each practice using random-effects meta-analysis.

In addition, we did a subgroup analysis to assess the RR of the intervention effect depending on individual practices' baseline performance. Practices were divided into two subgroups depending on whether their performance for each outcome was equal or below the pooled baseline and the RR was calculated for each subgroup.

### Referral rates

The data on referral numbers were based on National Hospital Episode Statistics extract data. Open Exeter data (a database of patient registration with National Health Service general practices)[32] were used to calculate the GP population list sizes.

### Data analysis

We used an interrupted time series (ITS) analysis to evaluate the impact of the intervention on referral rates. ITS is a powerful quasi-experimental design for evaluating effects of interventions when random assignment is not feasible[33] and is well suited to initial evaluations of community interventions with greater use of this method advocated for community intervention research.[34] This design was particularly useful for capitalising on existing data on GP referrals collected at CCG level. In order to limit the risk of bias, the quality criteria for ITS design as published by Effective Practice and Organisation of Care Cochrane group[35 36] were followed. Analysis using segmented time series regression technique (autoregressive integrated moving average-ARIMA) was used to control for baseline level and trend,[37] a common approach for evaluating policy and educational interventions.[38]

The analysis was undertaken using SPSS (V.22, IBM). Relative effects were calculated using the predicted values from the ARIMA analysis. The seasonality of the data was tested using a two-way analysis of variance (ANOVA). If the results showed significant monthly variation, seasonal decomposition was used (SPSS Statistics software) and the seasonally adjusted series (SAS) was entered in the ARIMA analysis.

### Referral rate variability

The referral rate variability among practices before, during and after the intervention was calculated from the musculoskeletal referral data. To compare the variability of referral rates before and after the intervention, the Levene's test was used.

### Patient experience

Patient experience with GP management was assessed via an anonymous questionnaire. The questionnaire collected information on whether the patient was examined, whether they had physiotherapy prior to the referral, and on the level of patient satisfaction (on a 5-point Likert scale) with the GP explanation and usefulness of their consultation.

### Data collection and analysis

We sent the first questionnaire survey to 130 consecutive patients attending the musculoskeletal community clinic a month prior to the intervention and the follow-up questionnaire survey was disseminated to an equal number of patients a month after the intervention. The data collection and analysis were undertaken by team members blind to the study design and procedures.

Comparisons of proportions between the groups were undertaken using the $X^2$ test or Fisher's exact test (categorical variables).

### Patient involvement

This was a professional intervention aiming to change GP behaviour. Patients were not involved directly in the design of the intervention. However, one of the main outcomes was patient experience because this is an important and often not adequately prioritised outcome when evaluating the effect of different referral management approaches.[19] The patient questionnaire incorporated questions from the General Practice Assessment Questionnaire and was piloted with a group of patients prior to its use.

## RESULTS
### Process evaluation

Clinical pathways for common musculoskeletal conditions were developed, actively disseminated to all practices and were available online. Monthly feedback on referral data comparing referral numbers at GP practice, network and borough levels was sent by the CCG to all GP practices and networks. The eight networks recruited local referral champions and all GP practices signed up to participate in the NIS. All practices sent representatives to at least 8 out of the 12 network-based educational meetings which aimed to discuss referral activity and best practice on referrals. The network-based meetings were facilitated by the referral champions and focused not just on referral activity but also on referral quality and the results of the clinical audits. Out of the 36 practices, one did not submit their clinical audit within the specified time frame while two submitted one of the two parts resulting in 33 (92%) complete returns.

Although a systematic evaluation of the mechanisms of change was outside the scope of this project, the end-of-year reports included practices' and networks' reflection on the process which gave some insight into how change might have occurred. Networks reported that the audits offered an opportunity for reflection on referral practice and identification of learning needs although two networks found the audits burdensome in terms of time. Practices and networks commented on the improved clinical knowledge and better awareness of local pathways and services which according to the theory of planned behaviour[39] and diffusion of innovation[40] can facilitate behavioural change. They also commented that feedback on referral rates, the clinical audits, the role of referral

**Table 2** Effect of intervention using relative risk (RR) on the five indicators of general practitioner referral quality

| | Effect of intervention | Pooled baseline proportion | Estimated postintervention proportion | Effect of intervention for practices with baseline proportion equal or below the pooled baseline proportion | Effect of intervention for practices with baseline proportion above the pooled baseline proportion |
|---|---|---|---|---|---|
| | RR (95% CI) | Proportion (%) (95% CI) | Proportion (%) (95% CI) | RR (95% CI) (n=no of practices) | RR (95% CI) (n=no of practices) |
| Adherence to pathways | 1.57 (1.35 to 1.81) | 42 (35 to 48) | 66 (57 to 76) | 2.09 (1.60 to 2.73) (n=17) | 1.38 (1.16 to 1.64) (n=16) |
| Letter information on suspected diagnosis | 1.31 (1.19 to 1.43) | 59 (53 to 65) | 77 (70 to 84 | 1.74 (1.47 to 2.05) (n=14) | 1.19 (1.09 to 1.29) (n=19) |
| Letter information on previous management | 1.21 (1.11 to 1.31) | 67 (61 to 73) | 81 (74 to 88 | 1.55 (1.35 to 1.78) (n=14) | 1.09 (1.02 to 1.15) (n=19) |
| Conversion of trauma and orthopaedics referrals to surgery | 1.45 (1.05 to 2) | 43 (34 to 52) | 62 (58 to 65 | 2.39 (1.56 to 3.67) (n=17) | 1.13 (0.87 to 1.46) (n=11) |
| Seen once and discharged | 0.51 (1.33 to 1.78) | 13 (10 to 17) | 6 (3 to 9 | 0.50 (0.30 to 0.80) (n=19) | 0.50 (0.16 to 1.52) (n=14) |

champions and participation in educational meetings created opportunities for shared learning and reflection which may have led to change via the social influence theory.[41] Finally, the teamwork and collaboration as demonstrated in the feedback comments may have contributed towards a culture of learning, an important component of successful QI initiatives.[42 43]

### Referral quality
Referral quality was assessed based on the information collected as part of the GP clinical audits. In total 521 musculoskeletal referral letters were audited and returned before the intervention and 463 after.

The clinical pathways covered the management of low back pain, shoulder pain and osteoarthritis of hip and knee joints. Half, 50% (259/521) of the referrals before the intervention and 49% (225/463) afterwards were for conditions covered in the above pathways. The results of these referrals were used to assess the effect of the intervention on pathway adherence.

Overall, there were statistically significant improvements across all five quality indicators (table 2). However, the effect of the intervention was markedly larger for practices with baseline performance equal or below the pooled baseline performance (table 2).

The above results are summarised in figure 1.

### Referral rates
Figure 2 presents the number of GP referrals to T&O and rheumatology specialties per 1000 patients per month between April 2009 and April 2016.

A two-way ANOVA showed significant seasonality of the data for T&O referrals, and therefore, the SAS was used for the analysis.

ARIMA analysis to establish whether there was an effect of the intervention revealed no statistically significant

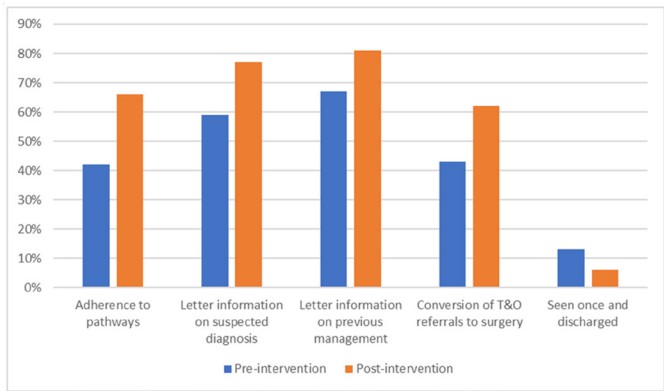

**Figure 1** Summary of audit results on the quality of GP referrals. GP, general practitioner; T&O, trauma and orthopaedics.

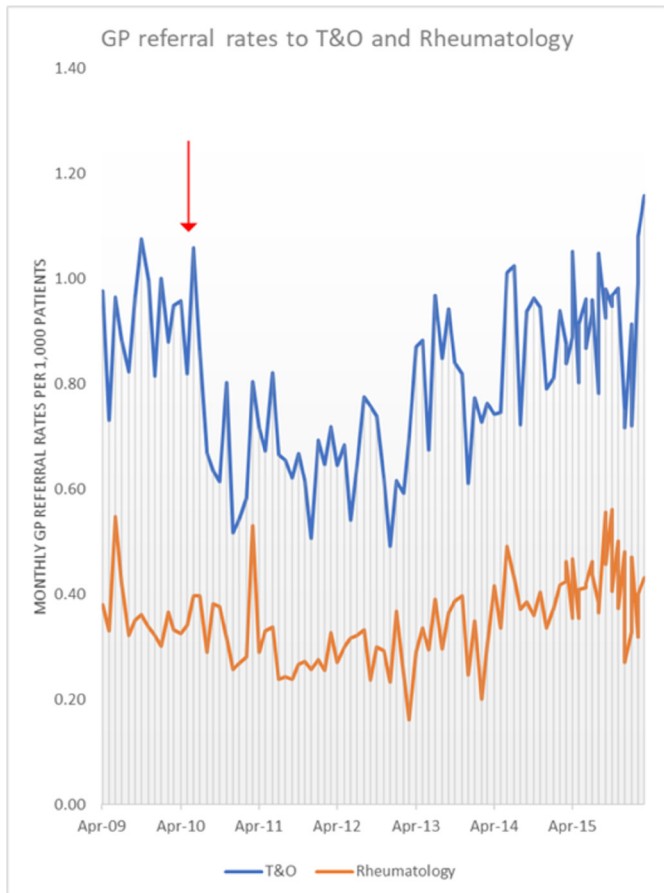

**Figure 2** Number of GP referrals to T&O and rheumatology per 1000 patients between April 2009 and April 2016 (the red arrow represents the time point of the intervention) GP, general practitioner; T&O, trauma and orthopaedics.

effect on rheumatology referral rates but there was a significant decrease in the T&O referral rates at 6, 12 and 18 months postintervention. The T&O results are summarised in table 3.

**Table 3** Effect of intervention on trauma and orthopaedics referral rates (using seasonality adjusted series ARIMA analysis)

|  | Estimate of effect | P value | 95% CI of estimate of effect | Relative effect (%) |
|---|---|---|---|---|
| Effect at 6 months | –0.31 | 0.001 | –0.48 to –0.14 | –32 |
| Effect at 12 months | –0.30 | 0.011 | –0.53 to –0.07 | –30 |
| Effect at 18 months | –0.30 | 0.048 | –0.59 to 0.00 | –30 |
| Effect at 24 months | –0.29 | 0.113 | –0.65 to 0.07 | –30 |
| Effect at 30 months | –0.28 | 0.191 | –0.71 to 0.14 | –28 |
| Effect at 36 months | –0.28 | 0.269 | –0.77 to 0.22 | –27 |

ARIMA, autoregressive integrated moving average.

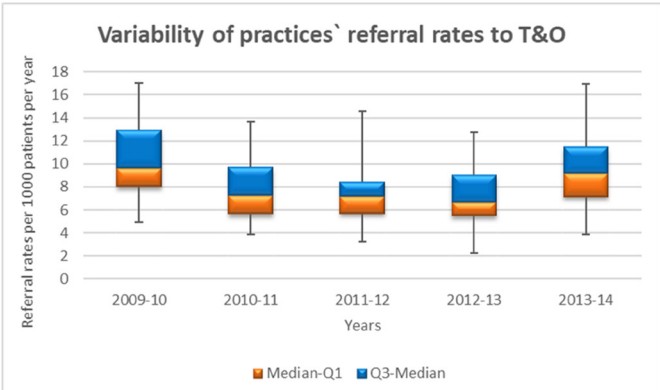

**Figure 3** Multiple box and whisker plot showing the variability of practices' referral rates to trauma and orthopaedics.

### Variability of referral rates among practices
The practice-level referral rates per 1000 patients per year followed a normal distribution (Shapiro-Wilk test >0.05, SPSS software). Reductions in the variability of referral rates among practices were shown in the 2 years following the intervention but these only reached statistical significance in the year after the intervention (2011–2012) for T&O referrals (Levene's test p=0.05) (figure 3).

### Patient experience
The response rate to the questionnaire was 81/130 (62%) prior to the intervention and 86/130 (66%) afterwards. All questionnaires returned were entered into the final analysis.

In terms of demographic characteristics, there were more participants from ethnic minority groups postintervention (exact $X^2$ for two independent proportions, p≤0.001).

More participants rated the GP explanation of their condition as good/very good postintervention (28/76 (37%) before vs 57/86 (66%) after, 95% CI 14% to 43%). Additionally, more participants rated the usefulness of their appointment positively (good/very good) postintervention (34/78 (43%) before vs 58/86 (67%) after, 95% CI 9% to 38%).

There were no changes with regard to examination rates and physiotherapy referrals before and after the intervention (74% (58/78) of the participants responded that they had been examined by the GP prior to the referral preintervention vs 77% (66/86) postintervention, (95% CI –10% to 16%) and 39/78 (50%) of participants responded that they had been referred for physiotherapy prior to their referral before vs 49/86 (57%) after the intervention (95% CI –8% to 22%)).

### DISCUSSION
The intervention resulted in improvements in the adherence of referrals to the clinical pathways, in the content of referral letters and in the conversion of T&O referrals to surgery. Referral rates were reduced at 6, 12 and

18 months following the introduction of the intervention. These results are consistent with the findings that Evans[44 45] reported in her evaluation of a referral improvement project using peer review in Torfaen and are particularly important in view of the nationally increasing trend of GP referrals to orthopaedics which was estimated by Briggs to be 7%–8% per annum, in his 2015 report entitled 'Getting it right first time'.[46]

Despite the observed actual reduction in rheumatology referral rates, this did not reach statistical significance according to the ITS analysis. This might be explained by the fact that the clinical pathways that were disseminated as part of the intervention mainly addressed conditions that were traditionally referred to T&O rather than rheumatology, such as radicular back pain and hip/knee osteoarthritis. Additionally, the simplicity of the message that patients who do not need or who would not consider surgery should not be referred to a surgeon, which was highlighted in the clinical pathways, may have assisted towards reducing T&O referrals. In contrast, referrals to rheumatology require more complex considerations, including assessing the possibility of inflammatory pathology, which are more challenging to address in a brief clinical pathway which may explain the smaller effect on the number of rheumatology referrals.

The improvements in the quality of referrals show that an intervention with a developmental focus, aiming to facilitate reflection on clinical practice and assist clinicians to identify their individual learning needs and find their own solutions can be effective. The NIS was a set of tools that acted as the enabler for clinicians to engage and was appropriate because maintenance of professional autonomy is important for GPs[47] and respect for professionalism has been shown to motivate them to engage in QI activities.[48]

It could be that the dialogue generated in the outreach meetings with the referral champions addressed some of the intrinsic psychological factors that are important determinants of GP referral behaviour. Such factors include reduced tolerance of uncertainty or a perception that serious disease is more frequent.[17]

Patient factors, for example, high patient expectations for a referral and anxiety about the condition also influence referral decisions.[18] The improvements recorded in patient experience are especially important in view of the fact that there was a statistically significant proportion of people from ethnic minority groups answering the postintervention questionnaire as there is some evidence that they tend to evaluate their care more negatively, even after analyses have been adjusted for potential confounders.[49 50] The questionnaire response rate compares very well with other questionnaire surveys in Tower Hamlets.[51]

The effect of the intervention was greater for practices with baseline performance below or at the pooled baseline. This is in accordance with a Cochrane systematic review assessing the effects of audit and feedback on the practice of healthcare providers and patient outcomes.[52]

Our findings suggest that it may be possible to increase the effect of interventions on a professional practice by optimally designing and better targeting these to achieve the maximum benefit at a reduced cost.

The lack of sustainability of the effect on referral rates could be explained by Bandura's social cognitive theory[53] which highlights the importance of stimuli and reinforcement for learning and behavioural change. Because of the turnover of clinical staff and the competing priorities within general practice, it may be important that components of the intervention which act as stimuli for reflection, such as the clinical audit and the facilitated peer-review meetings, are incorporated within the clinical practice for sustainable behavioural change.

The high engagement levels of GPs during the intervention are probably due to its developmental focus and its emphasis on improving the appropriateness and quality of referrals while respecting and empowering professional autonomy.[47 54] Our findings and some evidence that similar approaches may result in improvements in other geographical areas[15] and in different medical specialties[55] suggest that the intervention may be generalisable in other settings. As the healthcare system changes and becomes more integrated, the development of more patient-centred care pathways and the possibilities that derive from new technological developments present opportunities for building on professionalism and further improving referral processes through interprofessional cooperation.[56–58]

## Strengths and limitations

The intervention was implemented within the constraints of clinical commissioning and rapid policy change demonstrating its feasibility and applicability within similar health systems. Unlike other healthcare interventions, it was developed with an underpinning behavioural theory and analysis of its components indicated that it incorporated a number of behaviour influencing factors, including knowledge and skills, motivation and goals, social/professional role and identity, social influences and behavioural regulation.[24] Addressing the different barriers is important for the successful implementation of change.[59] The long follow-up on the impact on referral rates provided an opportunity to assess the sustainability of behavioural change over time.

One of the study's limitations was the fact that the design was by necessity non-randomised. The effect of the intervention may be partly attributable to increased attention, focus and priority on referral behaviour.[60]

A cost-effectiveness analysis would have been useful as such information can assist commissioning and health policy decisions on referral management interventions. Such decisions need to be based on cost-effectiveness data which will allow programme assessments to be made depending on the degree that interventions maximise health for the available resources and provide the highest 'value for money'. However, a cost-effectiveness analysis requires data from multiple sources in order to

allow an estimation of the impact to the whole health-care system rather than one part of it. It also requires data on patient-related outcomes which can be challenging to obtain as this is not routinely collected within primary care settings.

The intervention targeted behaviours at a number of levels and our evaluation could not identify the most cost-effective elements. There is conflicting evidence as to whether multifaceted interventions are more likely to be successful than single-component ones.[61 62] Most attempts to change professional behaviour involve bundles of interventions and because of their complex nature, it may not be possible to clearly assess the effectiveness of particular components.[63] A 2015 overview of systematic reviews suggested that 'interventions which contribute to normative restructuring of practice, modifying peer group norms and expectations (eg, educational outreach) and relational restructuring, reinforcing modified peer group norms (eg, reminders, audit and feedback), offer the best chances of success' and that 'combining such interventions is most likely to change behaviour'.[63] The concept of marginal gains has had a positive impact in sport and may offer a promising approach for enhancing healthcare professional behaviour and performance and for creating a culture of continuous improvement.[64]

## CONCLUSION

Our study shows that a QI-based approach to referral management which values GPs' professionalism can result in improvements across a range of outcomes including referral quality, patient experience, referral rates and variability. The intervention is feasible, well received by GPs and can be incorporated into every day clinical practice. Targeting the intervention where baseline performance is low may yield a greater effect.

**Author affiliations**

[1]Barts and The London School of Medicine and Dentistry, Queen Mary University of London, London, UK
[2]Warwick Medical School, Warwick University, Coventry, UK
[3]University Hospitals of Coventry and Warwickshire, Coventry, UK
[4]CHLS Central Office, Brunel University London, London, Uxbridge, United Kingdom

**Acknowledgements** The authors would like to thank Arthritis Research UK for funding this evaluation as part of an educational grant. We would also like to thank NHS Tower Hamlets Clinical Commissioning Group (CCG), the Clinical Academic Group (CEG-Centre of Primary Care and Public Health, Barts and The London School of Medicine and Dentistry) and the GPs in Tower Hamlets for their support and participation in the study.

**Contributors** All authors (VTB, MU, OMW and DM) designed the evaluation of the intervention. VTB led on the implementation, designed data collection tools, monitored data collection, wrote the statistical analysis plan, cleaned and analysed the data and drafted and revised the paper. She is guarantor. All remaining authors (MU, OMW and DM) contributed to the development of core ideas, the analysis plan, interpretation of the results and the drafting of the paper. NHS Tower Hamlets Clinical Commissioning Group (CCG) implemented and funded the intervention as part of the local network improvement scheme. The clinical pathways were developed in collaboration with the Clinical Academic Group (CEG), Centre of Primary Care and Public Health, Barts and The London School of Medicine and Dentistry.

**Funding** This work was supported by Arthritis Research UK (grant number 18678). Two of the authors are part-funded by the National Institute for Health Research (NIHR) (grant numbers GPPH1C6R and CATSCL-2013-04-003).

**Competing interests** All authors have completed the ICMJE uniform disclosure form at http://www.icmje.org/coi_disclosure.pdf. Arthritis Research UK has funded the submitted work. Dr Tzortziou Brown is part funded by the National Institute for Health Research (NIHR) (grant number GPPH1C6R). ProfessorUnderwood was chair of the NICE accreditation advisory committee until March2017 for which he received a fee. He is chief investigator or coinvestigator onmultiple previous and current research grants from the UK National Institutefor Health Research, Arthritis Research UK and is a coinvestigator on grantsfunded by the Australian NHMRC. He has received travel expenses for speaking atconferences from the professional organisations hosting the conferences. He isa director and shareholder of Clinvivo that provides electronic data collectionfor health services research. He is part of an academic partnership with Sercorelated to return to work initiatives. He is a coinvestigator on a studyreceiving support in kind from Orthospace. He is an editor of the NIHR journalseries for which he receives a fee. Professor Morrissey is part funded by the NIHR/HEE Senior Clinical Lecturer scheme (CAT SCL-2013-04-003). Professor Westwood declared no financial relationships with any organisations that might have an interest in the submitted work in the previous 3 years. All authors declared no other relationships or activities that could appear to have influenced the submitted work.

**Patient consent for publication** Not required.

**Ethics approval** The National Research Ethics Service (NRES) was contacted and confirmed in writing that ethical approval for the project was not required from the NHS research ethics committee (REC). Additionally, the Joint Research & Development office at Barts and The London was contacted and confirmed the same. Ethical approval was therefore granted by QMUL and permission to use the relevant data was also granted by the Governance Team of NHS North East London and The City.

**Provenance and peer review** Not commissioned; externally peer reviewed.

**Data sharing statement** The investigators will share data used in developing the results presented in this manuscript on request to the corresponding author.

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
