## [Reviewer comments · BMJ Open]

This paper was submitted to a another journal from BMJ but declined for publication following peer review. The authors addressed the reviewers' comments and submitted the revised paper to BMJ Open. The paper was subsequently accepted for publication at BMJ Open.

(This paper received three reviews from its previous journal but only two reviewers agreed to published their review.)

ARTICLE DETAILS

TITLE (PROVISIONAL)	Improving the management of musculoskeletal conditions: can an alternative approach to referral management underpinned by quality improvement and behavioural change theories offer a solution and a better patient experience? A mixed methods study
AUTHORS	Tzortziou Brown, Victoria; Underwood, Martin; Westwood, Olwyn; Morrissey, Dylan

VERSION 1 – REVIEW

REVIEWER	Simon Vulfsons Institute for Pain Medicine, Rambam Health Care Campus Technion, Israel institute of Technology, Haifa, Israel
REVIEW RETURNED	10-Jul-2018

GENERAL COMMENTS	This is a well thought out study, clearly defined and well executed. I do feel that more detail could be given on the intervention process. How were the interventions conducted, how many participants, how long etc. This is important for understanding and appreciating what exactly was done at these network based meetings.
--

REVIEWER	Trudy Rebbeck University of Sydney, Australia
REVIEW RETURNED	14-Sep-2018

GENERAL COMMENTS	Overall comments: The study is well described and robust in its methods. It demonstrates a change in GP behavior, that is difficult to achieve. Issues to address are as follows: Methods 1. Whilst the intervention (s) are well described, how adherence to best practice was measured is not. The manuscript could be improved if the authors explained how they evaluated referral practices. How did the authors consider referral to be appropriate? In what circumstances? What were the pathways to be adhered to ? How was adherence measured? 2. I could not see where ethics approval was obtained. I may have missed this, apologies if so.
---

	Results/ Discussion 3. There is no cost effectiveness or health care outcomes evaluated in this study. If the purpose of changing practice of behavior is to reduce cost and or improve health outcomes? Were these evaluated? If not why not? Please address this in the limitations. 4. As stated above, in results it would be good to see what exactly were the pathways adhered/ referral practices prior to and after the intervention? Eg less referral for MRI, to orthopedic surgeon, more appropriate referral – if so what were the indicators? Similar with the pathways—what was the ideal pathway? Where did behavior deviate from that pathway pre intervention and what aspects improved post intervention? The reader has no idea of this—just that overall pathway adherence improved. If the above can be addressed, the reader would be more informed.
--	--

REVIEWER	Sarah Dean University of Exeter Medical School, UK
REVIEW RETURNED	24-Oct-2018

GENERAL COMMENTS	This is a very nicely prepared manuscript and checklist, it was easy to follow and review, thank you. As I am not a statistician I would suggest this aspect of the work is reviewed by someone with such expertise. I have three points for the authors to consider including in a revised discussion and two recommended amendments regarding structure / presentation. 1) The work has excellent underpinning theory, use of behavioural change approaches and process / complex intervention design and the authors should be commended for integrating all this in a coherent way. I would however suggest expanding a little more on the comment on page 10, lines 9-12, regarding the lack of sustainability. Given that 'stimuli and reinforcement' were used during the intervention and GPs did appear to 'learn' (in that one year results were positive) I was less convinced by your explanation of lack of sustainability as being explained by Bandura's theory, surely if 'true' learning had occurred it should then be sustained? Or was the initial learning not sufficiently embedded to persist over time? Bandura's theory does help us understand the initial changes that occurred but does it help explain the lack of sustainability? Do you think the intervention needed to be carried out for longer (but then for how long do we need to repeat the stimuli and reinforcements)? Or instead do we need to implement better maintenance / relapse recovery strategies? 2) It would be worth adding some comment as to why the changes happened for Trauma & Orthopaedics referral rates but not for Rheumatology. Do you think it was 'easier' for GPs to improve their decision making for likely orthopaedic surgery versus likely rheumatology intervention? Were there any NHS contextual issues impacting at the time of the study that might also explain these results (for example GPs reduced referrals because they knew elective surgery was being put on hold over the winter? – see next point too). Please note I have not checked whether this NHS context did occur locally at the time of your study but I believe it would be worthwhile making it clear that this was - or was not - the case.
--

	3) It would be useful to know if the plan for seasonal adjustment noted in the data analysis section (page 6 lines 10-11) was undertaken. If it was used then I can see that my concern above would be redundant, however I am not able to comment on the statistical techniques chosen for seasonal adjustment. 4) I was surprised that the manuscript ended with the 'Strengths and Limitations' section, would it be worthwhile adding a 'Conclusion'? 5) Something has gone wrong with the reference list; in my printed version of your manuscript there are no journal names. Overall a very interesting and useful report and I wish the authors well with their future research and implementation work.
--	--

VERSION 1 – AUTHOR RESPONSE

Reviewer: 1

This is a well thought out study, clearly defined and well executed.

We thank the reviewer for their positive appreciation of our work.

I do feel that more detail could be given on the intervention process. How were the interventions conducted, how many participants, how long etc. This is important for understanding and appreciating what exactly was done at these network-based meetings.

The reviewer is right. Additional information has now been added both within Table 1 which summarises the components of the intervention and in the next paragraph of the main text. The components of the intervention within the table now read:

Local consensus processes: Local clinical pathways on common musculoskeletal conditions (low back pain, shoulder pain and osteoarthritis) designed with input from multidisciplinary (MDT) teams (using the Delphi technique) summarizing guideline recommendations and availability of local services. These pathways were disseminated to all 36 practices, were presented at local educational events and were published on the local website for GPs to access.

Clinical audit: Audit and reflection on current practice using standardised proformas, opportunity for team discussion of findings, identification of learning needs and agreeing ways to improve practice. The audits took place at practice level and the results were discussed both within practices and also at network events.

Feedback: Monthly reports to provide ongoing feedback to clinicians on referral activity (comparative data at GP, practice, network and borough levels). These reports were disseminated both at practice and network levels.

Monthly educational meetings and outreach visits facilitated via local opinion leaders: Opportunity to discuss the clinical pathways and clinical practice with peers (case-based discussions). Local opinion leaders (referral champions) in each network of practices facilitated discussions on referral decision making, best clinical practice and identification of learning needs. The meetings were attended by GPs and practice managers from each practice.

The paragraph that follows Table 1 now reads:

The intervention was part of the Tower Hamlets Clinical Commissioning Group (CCG) a Network Improvement Scheme (NIS), which acted as the system driver providing the incentives for change while ensuring a rigorous, structured approach incorporating planned objectives within set timelines over a year's period. The CCG's project management team ensured that the intervention would be delivered at scale and pace. The NIS offered with allocated funding at network level for the clinical audits, for hosting educational events and for clinician backfill to attend these. There were no financial incentives attached to clinical performance targets such as referral rates because such incentives may have a negative impact on the quality of care (29) and introduce conflicts of interests and ethical dilemmas.

(30)

Reviewer: 2

Overall comments:

The study is well described and robust in its methods. It demonstrates a change in GP behavior, that is difficult to achieve.

We thank the reviewer for their positive comments.

Issues to address are as follows:

Methods

1. Whilst the intervention (s) are well described, how adherence to best practice was measured is not. The manuscript could be improved if the authors explained how they evaluated referral practices. How did the authors consider referral to be appropriate? In what circumstances? What were the pathways to be adhered to? How was adherence measured?

We agree with the reviewer that some more information on how adherence to the pathways was measured would be helpful. Therefore, the relevant paragraph on Referral quality (under Outcomes evaluation) has been changed to read:

To assess the impact on referral quality, we assessed three attributes as described by Blundell et al (31): necessity, quality of process, and destination. To determine necessity, we measured the adherence of referrals to the agreed local clinical pathways. on back pain, shoulder or osteoarthritis (provided that the reason for the referral was one of the above conditions). The pathways covered all the aspects of management within primary care including the indications for investigations, medication, physiotherapy, the role of steroid injections and provided resources on exercise and advice for patients. The information on the referral letter and EMIS notes was compared with the relevant pathway during the audit process. The EMIS records were used in addition to the information contained in the referral letters in order to ensure that all relevant information on the prior management of patients was captured. We recognised that a simple yes/no answer could be subjective and might not reliably capture adherence to the clinical pathway. Therefore, the audit proforma incorporated questions on the specific management of the patient prior to the referral and on what could have been done differently according to the pathway. We used the information contained in the referral letters as an indicator of quality of process. Finally, we calculated surgical conversion rates and the percentage of referrals that were seen once and discharged from the hospital without intervention and used these as indicators of the appropriateness of referral destination.

2. I could not see where ethics approval was obtained. I may have missed this, apologies if so.

We thank the reviewer for this comment. We have incorporated information on ethics approval within the submission documentation which reads:

The National Research Ethics Service (NRES) was contacted and confirmed in writing that ethical approval for the project was not required from the NHS research ethics committee (REC). Additionally, the Joint Research & Development office at Barts and The London was contacted and confirmed the same. Ethical approval was therefore granted by QMUL and permission to use the relevant data was also granted by the Governance Team of NHS North East London and The City.

Results/ Discussion

3. There is no cost effectiveness or health care outcomes evaluated in this study. If the purpose of changing practice of behavior is to reduce cost and or improve health outcomes? Were these evaluated? If not why not? Please address this in the limitations.

We agree with the reviewer that a cost-effectiveness analysis and an assessment of the impact of the intervention on patient outcomes would have been helpful. However, undertaking a detailed cost-effectiveness analysis was beyond the scope of this project due to lack of relevant data and resources. We considered other primary outcomes including patient related outcomes such as patient function and quality of life. However, although there are validated tools for the measurement of such outcomes, these

are not consistently measured at a primary care level. Because of the above, the quality of GP referrals was chosen as the main primary outcome. We have added the following paragraph under the limitations of the study as advised by the reviewer:

A cost-effectiveness analysis would have been useful as such information can assist commissioning and health policy decisions on referral management interventions. Such decisions need to be based on cost-effectiveness data which will allow programme assessments to be made depending on the degree that interventions maximize health for the available resources and provide the highest 'value for money'. However, a cost-effectiveness analysis requires data from multiple sources in order to allow an estimation of the impact to the whole health care system rather than one part of it. It also requires data on patient-related outcomes which can be challenging to obtain as this is not routinely collected within primary care settings.

4. As stated above, in results it would be good to see what exactly were the pathways adhered/ referral practices prior to and after the intervention? Eg less referral for MRI, to orthopedic surgeon, more appropriate referral – if so what were the indicators? Similar with the pathways—what was the ideal pathway? Where did behavior deviate from that pathway pre-intervention and what aspects improved post intervention? The reader has no idea of this—just that overall pathway adherence improved. If the above can be addressed, the reader would be more informed.

We thank the reviewer for this comment. As we mentioned in the manuscript, to assess the impact on referral quality, we assessed three attributes as described by Blundell et al: necessity, quality of process, and destination. To determine necessity, we measured the adherence of referrals to the agreed local clinical pathways on back pain, shoulder or osteoarthritis (provided that the reason for the referral was one of the above conditions). The pathways covered all the aspects of management within primary care including the indications for investigations, medication, physiotherapy, the role of steroid injections and provided resources on exercise and advice for patients. We used the information contained in the referral letters as an indicator of quality of process. Finally, we calculated surgical conversion rates and the percentage of referrals that were seen once and discharged from the hospital without intervention and used these as indicators of the appropriateness of referral destination.

Although the clinical audits elicited some specific information on the management of patients prior to referral, a detailed analysis of the differences in specific aspects of the patient management (e.g. prescription rates, investigation requests, physiotherapy referral rates and steroid injections) before and after the intervention was outside the scope of the project. However, we did report on the impact of the intervention on referral rates to T&O and rheumatology and on the outcomes of such referrals (surgical conversion rates and seen once and discharged) which give an indication on appropriateness.

Reviewer: 3

This is a very nicely prepared manuscript and checklist, it was easy to follow and review, thank you.

We thank the reviewer for these encouraging comments.

As I am not a statistician I would suggest this aspect of the work is reviewed by someone with such expertise.

I have three points for the authors to consider including in a revised discussion and two recommended amendments regarding structure / presentation.

1) The work has excellent underpinning theory, use of behavioural change approaches and process / complex intervention design and the authors should be commended for integrating all this in a coherent way. I would however suggest expanding a little more on the comment on page 10, lines 9-12, regarding the lack of sustainability. Given that 'stimuli and reinforcement' were used during the intervention and GPs did appear to 'learn' (in that one year results were positive) I was less convinced by your explanation of lack of sustainability as being explained by Bandura's theory, surely if 'true' learning had occurred it should then be sustained? Or was the initial learning not sufficiently embedded to persist over time? Bandura's theory does help us understand the initial changes that occurred but does it help explain the lack of sustainability? Do you think the intervention needed to be carried out for longer (but

then for how long do we need to repeat the stimuli and reinforcements)? Or instead do we need to implement better maintenance / relapse recovery strategies?

We thank the reviewer for this constructive feedback. We have changed the relevant paragraph on page 10 to read:

The lack of sustainability of the effect on referral rates could be explained by Bandura`s Social Cognitive Theory (54) which highlights the importance of stimuli and reinforcement for learning and behavioural change. Because of the turnover of clinical staff and the competing priorities within general practice it may be important that components of the intervention which act as stimuli for reflection, such as the clinical audit and the facilitated peer-review meetings, are incorporated within clinical practice for sustainable behaviour change.

2) It would be worth adding some comment as to why the changes happened for Trauma & Orthopaedics referral rates but not for Rheumatology. Do you think it was 'easier' for GPs to improve their decision making for likely orthopaedic surgery versus likely rheumatology intervention? Were there any NHS contextual issues impacting at the time of the study that might also explain these results (for example GPs reduced referrals because they knew elective surgery was being put on hold over the winter? – see next point too). Please note I have not checked whether this NHS context did occur locally at the time of your study but I believe it would be worthwhile making it clear that this was - or was not - the case.

We thank the reviewer for this comment and we agree that the possible reasons for the differential effect of the intervention on rheumatology referral rates need to be mentioned in the discussion section. We have therefore added the following paragraph:

Despite the observed actual reduction in rheumatology referral rates, this did not reach statistical significance according to the interrupted time series analysis. This might be explained by the fact that the clinical pathways that were disseminated as part of the intervention mainly addressed conditions that were traditionally referred to T&O rather than rheumatology, such as radicular back pain and hip/knee osteoarthritis. Additionally, the simplicity of the message that patients who do not need or who would not consider surgery should not be referred to a surgeon, which was highlighted in the clinical pathways, may have assisted towards reducing T&O referrals. In contrast, referrals to rheumatology require more complex considerations, including assessing the possibility of inflammatory pathology, which are more challenging to address in a brief clinical pathway which may explain the smaller effect on the number of rheumatology referrals.

3) It would be useful to know if the plan for seasonal adjustment noted in the data analysis section (page 6 lines 10-11) was undertaken. If it was used then I can see that my concern above would be redundant, however I am not able to comment on the statistical techniques chosen for seasonal adjustment.

We thank the reviewer for the request for this clarification. The seasonality of the data was tested using two-way ANOVA and showed significant seasonality for T&O referrals. The seasonality of the data was adjusted using seasonal decomposition (SPSS) and we used the seasonally adjusted series for the analysis. We have incorporated the following sentence in the results section:

Two-way ANOVA showed significant seasonality of the data for T&O referrals and therefore the seasonally adjusted series (SAS) was used for the analysis.

4) I was surprised that the manuscript ended with the 'Strengths and Limitations' section, would it be worthwhile adding a 'Conclusion'?

We agree with the reviewer that ending with a conclusion may be more helpful. We have added the following conclusion at the end of the manuscript:

Our study shows that a quality improvement-based approach to referral management which values GPs` professionalism can result in improvements across a range of outcomes including referral quality, patient experience, referral rates and variability. The intervention is feasible, well received by GPs and can be incorporated into every day clinical practice. Targeting the intervention where baseline performance is low may yield a greater effect.

5) Something has gone wrong with the reference list; in my printed version of your manuscript there are no journal names.

We thank the reviewer for bringing this to our attention. We have revised the reference list and ensured this is now according to the BMJ Open guidance and it includes the journal names.

Overall a very interesting and useful report and I wish the authors well with their future research and implementation work.

We are grateful to the reviewer for these positive comments.

VERSION 2 – REVIEW

REVIEWER	Sarah Dean University of Exeter, UK
REVIEW RETURNED	19-Dec-2018
GENERAL COMMENTS	Thank you for providing a clear list of revisions.